

# Brief communication: Electron pair donors and Earth's energy generation

Frederick Mayer

Mayer Applied Research Inc.,
1417 Dicken Drive
Ann Arbor, Michigan 48103

*Correspondence to:* Frederick Mayer (fmayer@sysmatrix.net)

**Abstract.** This Brief Communication presents a series of model calculations for the electron pair donor densities required for *tresino* thermal energy generation in the Earth. The crucial density of electron donors is determined from the ratio of $He^3$ and $He^4$ after many years starting from initial densities of the donor pairs. In addition, a new proposal is introduced that connects Cooper pair formations to the deuteron *tresino* nuclear reaction chain (the chain that determines the $He^3/He^4$ ratio).
Furthermore, it is proposed that magnetotelluric (MT) observations may be connected to Cooper pair formation either with or without substantial heating.

## 1   Electron-Pair Donors and *Tresino* Formation

This Brief Communication describes a new proposal relating to *tresino* formation–hence to thermal energy generation (Mayer and Reitz, 2014) in the Earth; it represents an extension of our prior work on the thermal energy generation in the Earth. In our
NPG paper, we made an assumption regarding the collisions that gave rise to the transfer of a pair of electrons required to form a *tresino*, hence energy generation. The assumption was that the electron pairs were delivered in a collision between a proton (probably $H_3O^+$ i.e., a hydronium ion) and a doubly negatively-charged ion, for example $O^{2-}$ (see Panel A of Figure 1), in which the weakly-bound electrons were captured in the collision. In this paper, an alternative process is suggested, namely: that, in the Earth's very mixed and varied materials of both insulators and metals, a "superfluid" of Cooper electron pairs forms
in some regions such that the pairs can migrate and eventually collide with a proton (here again, probably $H_3O^+$ or its deuteron cousin) (see Panel B of Figure 1) to form either a proton *tresino* or a deuteron *tresino*. The ensuing reaction dynamics and energy generation then follows the same reaction chains as those of our earlier paper (Mayer and Reitz, 2014). Unfortunately, the microphysics of the formation of the Cooper pairs is itself complex because of the physical processes, the materials, and the spatial length scales may all be diverse even in laboratory experiments, which by the way, are generally done at low
temperatures, as described in the overview paper of (Hirsh, Maple and Marsiglio , 2015).

Cooper pairs have been recently been proposed in Feigel'man and Ioffe (2015) and S. Dolgopolov (2015) in somewhat mixed materials, including at interfaces, see e.g.[Gariglio, et.al. (2015)]. So considering "superfluids" of Cooper pairs created in Earth materials, perhaps under pressure, seems a reasonable assumption in the Earth. Of course, assessing the materials most operative in the Earth will have to be determined. Interestingly, the "superfluid" has only to be a (local) transient process but



repeated frequently for the Cooper pairs to exist long enough to collide with a proton or deuteron. So, this is the *new* proposal: Cooper pairs form in the Earth materials in some situations and if a sufficient amount of acidified water is present, they can form *tresinos* in collisions, thereby driving the sequence of reactions that we described in (Mayer and Reitz, 2014).

Note that in many Cooper pair theories the spins of the two electrons are opposed which is just the situation required for the pair to "fall into" the proton potential well forming a *tresino*, therefore releasing its binding energy as recoil kinetic energy. This collision is illustrated in Panel B in Figure 1; the broken circles in Figure 1 surrounding composite particles is meant to indicate that they are bound. Further, note that the binding energy released at *tresino* formation is delivered as recoil kinetic energy ($\approx 3.7$ keV) to the participating *tresino* and its heavier partner, in both cases, a water molecule. It is important to point out that a "superfluid" of Cooper pairs may already have been observed for many decades in magnetotelluric (MT) scans but just not recognized as such.

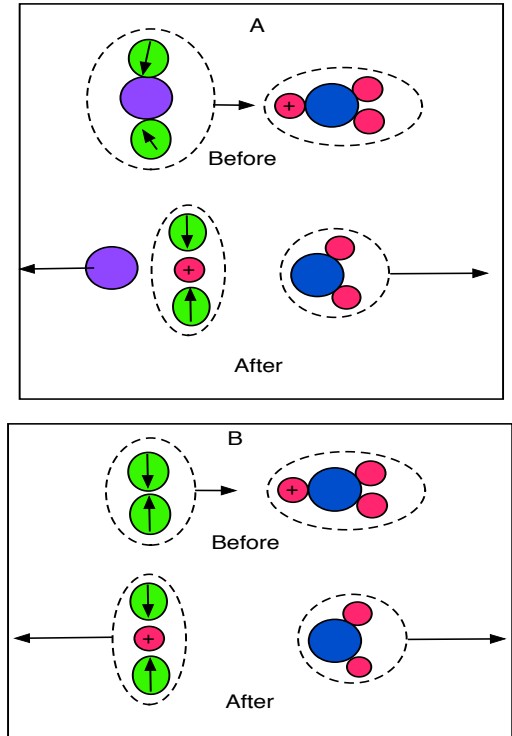

**Figure 1.** Panel A: schematic of electron capture by an O$^{2-}$ ion. Panel B: schematic of electron capture by a Cooper pair. Electrons in green with spin orientations as shown, protons in red and oxygen atoms in blue.

10



## 2 The Ratio of He$^3$ and He$^4$

It has proved useful to examine a series model calculations of the helium isotope ratio because this ratio can be traced back to the amount of electron-pair donor density at the start of the *tresino* energy release. It has been instructive to use the *tresino* reaction dynamics of our paper (Mayer and Reitz, 2014) to examine the effect of differing levels of water for proton and

5 deuteron *tresino* formation that generate the $^3$He and $^4$He isotopes. In particular, the deuteron *tresino* nuclear reaction chain that results in the level of the helium isotopes produced, is sensitive to the starting value of the electron-pair donor density. And the ratio of $^3$He and $^4$He isotopes of these two isotopes has been well-documented as we discussed in (Mayer and Reitz, 2014). The long-time (beyond about 200 years) ratio and far away from any volcanic activity, has been found to be $\approx 10^{-5}$ around the Earth. Therefore, by adjusting the amount of water (i.e., proton and deuteron densities) it is possible to estimate the

10 amount of water content as well as the donor density of electrons either from ($O^{2-}$) ions in Panel A or from Cooper pairs in Panel B. A sequence of model calculations is shown in the upper panel of Figure 2. The values taken for these calculations was [0.5, 1, 2, 3, 4, 5 and 6] $\times 10^{20}$ protons/cm$^3$ at the start of an approximately one year introduction of water ($\nu = 1$). The curve yielding the isotope ratio curve at $6 \times 10^{20}$ protons/cm$^3$ appears to agree best with the geophysically-observed ratio data beyond about 200 years. So, this data point also provides an estimate of the Cooper pair density at the same value $6 \times 10^{20}$ pairs/cc. On

the other hand, a similar density of doubly-charged negative ions such as $O^{2-}$ as we had suggested in our earlier NPG paper gives this same result. Finally, note the linear dependence of the power output verses the proton content as shown in the lower panel of Figure 2. So, with very little water there is very little energy generation. Of course, the model calculations does not select one or the other of these electron pair donor possibilities but they do indicate the densities required in the *tresino* energy generation picture.

## 3 Superfluids and Magnetotelluric Data

Magnetotelluric data scans often find zones of very-high electrical conductivity; see, for example, a typical paper by Ritter, et al. (1999). Furthermore, as these authors comment "A large number of electrical conductivity anomalies have been detected in the Earth crust around the world. There is no clear consensus as to the causes and origins of these anomalies, particularly in crystalline regimes". On the other hand, it is clear from basic electromagnetic considerations that these zones have enhanced

electrical conductivity, a situation certainly consistent with generation of Cooper pairs. It is important to note that such zones may be found even without substantial heating (from *tresino* generation) in those situations where there is insufficient acidified water (hydronium ions) present, as well as lower levels of heating from small amounts of acidified water. It should be obvious that there must be equal numbers of hydronium ions for each Cooper pair to undergo a *tresino* formation in order to release its binding energy.

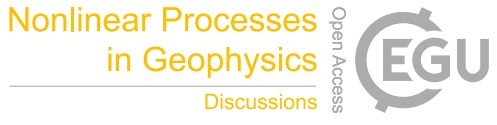

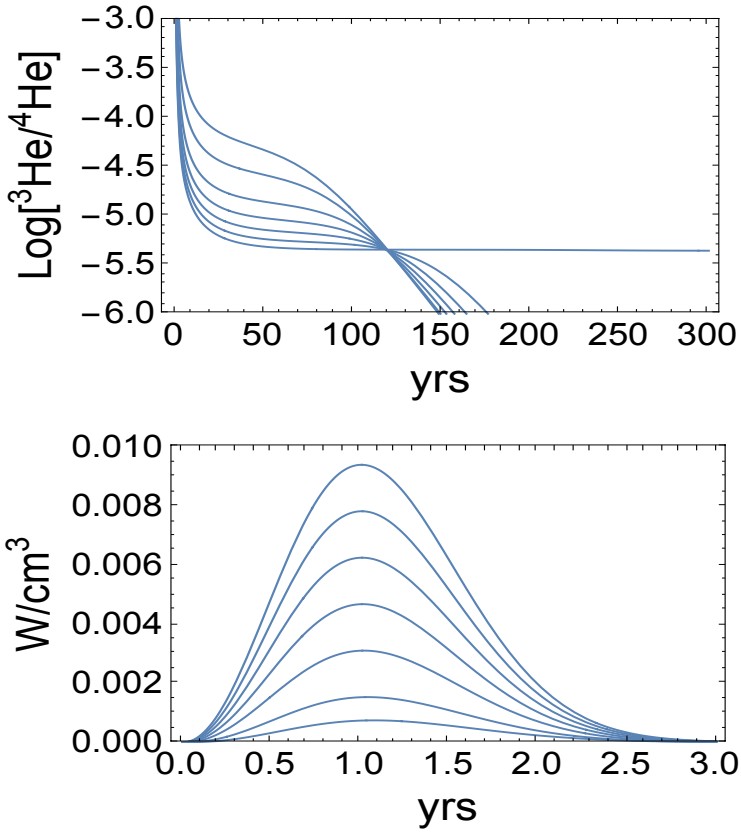

**Figure 2.** The upper panel plots the helium isotope ratios for different amounts of water or starting proton content: [0.5, 1, 2, 3, 4, 5, and 6]$\times 10^{20}$ protons/cm$^3$ at t=0. Other relevant parameters (defined in our earlier NPG paper) for the model calculations were: $\nu = 1$, $T = 1555°K$, $P = 2.33 \times 10^{-5}$, $n_d(0) = n_p(0)/6600$, $\eta = 10^{-16}$, $\epsilon = 0$, and $n_{ee}(0) = 2n_p(0)$. The lower panel plots the power output (W/cm$^3$) in early times for the same range of starting proton densities.

## 4 Discussion

There does not seem to be a straightforward way to distinguish between the two different processes by which a *tresino* can be formed either: (a) a collision between a hydronium ion and a doubly-negatively charged ion such as O$^{2-}$ and (b) a collision between a hydronium ion and a Cooper pair. Of course, much of the research into superconducting materials (Herrman & Maple, 1991) has focussed upon low-temperature systems even though much research has pushed on to some higher-temperature superconductivity materials due to their practical importance. However, the existence of HTSC suggests that even higher-temperature superconductivity might be found in geophysical systems but not yet examined.



## 5    Conclusions

Of course, this new proposal will have to be looked at closely to determine if there may be geophysical data that supports it (or not) and also what geochemical components there may be that might allow it to be examined in the laboratory. Finally, if those geochemical components can be identified, then a new research direction for understanding "superconductivity" may present

5    itself.





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

*Competing interests.* There are no competing interests in this paper.

*Acknowledgements.* With gratitude, I acknowledge my late colleague, mentor, and friend, Dr. John Reitz, without whom this work would never have become possible.