# Peer review of "Brief communication: Electron pair donors and Earth's energy generation"

_Nonlinear Processes in Geophysics, 2018_

## Author Comment (AC1) · 30 Mar 2018

Author's Comment:

Just after completing this Brief Communication, the author became aware of some significant new research that may represent the basis of the supercurrents (Cooper pairs) in the geophysics setting. Specifically, new research (1,2) indicates that slightly misaligned planes in graphene (in graphite) may produce the pairs. Furthermore, graphite is certainly a common geophysically significant component in some geophysical situations, e.g. (3). In addition, water is easily intercalated in graphitic structures (4) and, importantly, has attracted attention as a possible room ■temperature superconductor. So, the two important components required for delivering the superconducting pairs

are therefore in position for producing tresinos and therefore energy generation and magnetotelluric measurements.

(1) see "Unconventional superconductivity in graphene bilayers", the article by Johanna L. Miller in 22 March (2018) Physics Today Research & Technology.

(2) "Magic-angle graphene superlattices: a new platform for unconventional superconductivity", Yuan Cao, Valla Fatemi, Shiang Fang, Kenji Watanabe, Takashi Taniguchi, Efthimos Kaxiras, and Pablo Jarillo-Herrero, arXiv:1803.02342v1 [cond-mat.mes-hall] 6 Mar (2018).

(3) "Microscopic scale conductivity as explanation of magnetotelluric results from the Alps of Western Switzerland", Gabriella Losito, Pierre-André Schnegg, Candice Lambelet, Cecilia Viti, Antonello Trova, Geophysical Journal International, Volume 147, Issue 3, 1 December (2001), Pages 602–609.

(4) "Theoretical study of graphite intercalated with water cyclic hexamers", R.M. Torres Rojas, R. Baquero, Carbon, V107, October (2016), Pages 332 -337

---

## Referee Comment (RC1) · S. Gariglio (Referee) · 13 Apr 2018

The manuscript "Electron pair donors and Earth's energy generation" submitted by F. Mayer for publication in Nonlin. Processes Geophys. Discuss. develops further the idea published in a previous paper (Mayer and Reitz, 2014) of energy generation by the tresino nuclear reaction chain. The current work focuses on the source of the electron pairs and their amount to justify the long term ratio of He isotopes. I found the proposal interesting but the paper requires major revision before publication.

The output of a series of model calculations is plot in Figure 2 and, to my understanding, is the main scientific information the paper is providing. It would be important to show the behavior of the long term He isotopes ratio for proton content larger than

6x10ˆ20 cmˆ-3. It would also help to have different colors for the different proton content for the two plots in the Figure.

The proposal of Cooper pairs as source of electron pairs for the tresino nuclear reaction is interesting. To my knowledge, there are no experimental evidences of their existence inside the Earth but that would be indeed a wonderful discovery! There is one point the author should consider in the manuscript. In order to have Cooper pairs, electrons need an attractive interaction that overcomes their natural Coulomb repulsion. In standard superconductors, this interaction is due to phonons, i.e. the vibrations of the crystal lattice; in novel and exotic superconductors, it is thought to derive from magnetic fluctuations. A lot of research is ongoing to clarify this point on these new superconductors but so far, superconductivity manifests at low temperatures. What would be the mechanism that could lead to high temperature Cooper pair formation inside the Earth? Despite the author states that it is difficult to distinguish between the possible sources of electron pairs, the proposal would be more strong developing this point.

There are also few issues (style/format) that should be revised.

The introduction is not clear unless the reader knows the previous paper (Mayer and Reitz, 2014): it would be better to summarize its main findings in the current manuscript before moving on to present the new proposition.

Introduction, page 1 line 21: "Cooper pairs have been recently been proposed .." -> "Cooper pairs have recently been proposed .."

The sentence "Of course, assessing the materials most operative in the Earth will have to be determined." is not clear. Does it mean that superconductivity will be only present in certain rocks? Does the nuclear reaction occurs deep inside the Earth or at its surface?

Introduction page 2 line 6: "the broken circles ...is meant " -> "the broken circles ... are meant ";

Introduction page 2 line 9: a reference should be added to the citation of MT scans;

Figure 1: adding the labels to the "molecules" in the Figure could help the reader;

Page 3 line 2 : "a series model calculations" -> "a series of model calculations";

Page 3 line 7 "And the ratio of 3He and 4HE isotopes of these two isotopes.." : remove "of these two isotopes";

page 3 line 16: "verses" -> "versus".

---

## Author Comment (AC2) · 21 Apr 2018

Author's General response to s. Gariglio (Referee 1):

First, the Author wants to thank Referee 1 for his close reading of, and suggestions for, my "Brief communication: Electron pair donors and Earth's energy generation". These comments have helped make the presentation clearer. Furthermore, I have tried to make all the changes that reflect the issues Referee 1 has brought-up. Except for trying to argue details of the Cooper pair microphysics. Here, I referenced the Referee 1's paper and some in my Author's Comment that points to some possibilities. Unfortunately, there is no obvious mechanism for the Cooper pair formations in the geophysical setting. The requirement for some process leading to an electron pair

being donated in the tresino energy generation according to quantitative success of our earlier NPG (Mayer and Reitz, 2014) paper has made the Cooper pair a reasonable alternative choice.

Author's Specific response:

"The output of a series of model calculations is plot in Figure 2 and, to my understanding, is the main scientific information the paper is providing". This is correct except for the possibility of Cooper pairs playing an energy generation role.

Author's Specific response:

"It would be important to show the behavior of the long term He isotopes ratio for proton content larger than paper 6x10ËĘ20 cmËĘ-3." It turns out that above this particular value, the density of deuteron tresinos has gone to zero so no further increase is possible with the given choice of the other model parameters. Note here an important point: This is likely to be the reason that no geophysical data are found with a higher value of 3He/4He.

Author's Specific response:

"It would also help to have different colors for the different proton content for the two plots in the Figure." This suggestion by the Referee has been adopted in my revised paper.

Author's Specific response:

"To my knowledge, there are no experimental evidences of their existence inside the Earth but that would be indeed a wonderful discovery!" I agree with this Referee's comment.

Author's Specific response:

"Of course, assessing the materials most operative in the Earth will have to be determined." It seems that some class of materials may be responsible, and yes my guess

is that superconductivity will be only present in certain types of geophysical materials."

"Does the nuclear reaction occurs deep inside the Earth or at its surface?" In our earlier NPG paper (Mayer and Reitz, 2014), we had determined that the deuteron nuclear chain reactions take place relatively close to the surface rather than deeper in the Earth.

Author's Specific response:

"There is one point the author should consider in the manuscript. In order to have Cooper pairs, electrons need an attractive interaction that overcomes their natural Coulomb repulsion. In standard superconductors, this interaction is due to phonons, i.e. the vibrations of the crystal lattice; in novel and exotic superconductors, it is thought to derive from magnetic fluctuations." I agree with the Referee's comment here too. But the Author doesn't have anything further to contribute to this discussion, the Referee's discussion makes the physics points clearly. I do mention in the paper that the microphysics of Cooper pairs is undergoing much research as Referee 1 is well-aware of.

Author's Specific response:

"The introduction is not clear unless the reader knows the previous paper (Mayer and Reitz, 2014): it would be better to summarize its main findings in the current manuscript before moving on to present the new proposition". The Author has added a new Introduction section as per the Referee's suggestion.

---

## Referee Comment (RC2) · Anonymous Referee #2 · 8 May 2018

In the manuscript on "Electron pair donors and Earth's energy Generation" the author F. Mayer discusses an unusual concept for thermal energy generation in the earth based on "tresinos" and "Cooper pair formations" and he furthermore suggests that zones of high electrical conductivity found with magnetotelluric measurements may be caused by these.

The paper of Ritter et al. (1999) is cited out of context. It describes a significant electrical conductivity anomaly found beneath the Münchberger Gneismasse in Germany, which was interpreted to be a remnant of past tectonic processes. The high conductivity was attributed to graphite along shear planes. Graphitization was a plausible explanation for the highly conductive material because (i) graphite forms interconnected networks over large distances, necessary to be detectable with magnetotellurics, (ii)

[Figure]

graphite remains in place and stable over geological time spans (millions of years), which is necessary in the absence of active tectonic processes, (iii) graphite has a self-lubricating effect, thereby facilitating movement along shear planes, and (iv) graphite has been found in other large fossil shear zones around the world. In active tectonic regimes and in general fluids (including for instance partial melts) play a dominant role in explaining high electrical conductivity within earth. In fact, a multitude of papers exist on the interpretation of high electrical conductivity, which favour much simpler and more coherent conductivity mechanisms and which are supported by data and observations. None of these are mentioned or cited by the author.

I cannot comment on the soundness of the concepts of "tresinos" and "Cooper pair formations" as this is not my field of expertise and these concepts are not explained in the manuscript. The connection to electrical conductivity within the earth's deep interior, however, is not substantiated, very speculative, and not supported by any data. The only reference to a paper on magnetotellurics is cited out of context. Important references are missing, I therefore strongly advise against publishing this manuscript.

---

## Author Comment (AC3) · 9 May 2018

Anonymous Referee #2 has raised a number of criticisms and suggests rejecting my Brief Communication. I respond to his criticisms below.

First, Referee #2 mentions "an unusual concept for thermal energy generation in the earth based upon "tresinos" and Cooper pair formations". I think that unusual is an inappropriate description of the "tresino" energy generation process; unconventional might have been more appropriate as the geophysics community generally continues to believe in the Standard Earth Energy Paradigm (SEEP) that has been discredited in our earlier NPG paper (Mayer and Reitz, 2014). But, if Referee #2 had taken the time to read and understand our earlier NPG paper, he might have seen that the "tresino"

Earth energy generation picture resolves the numerous (observational and theoretical) paradoxes of the SEEP.

Second, Referee #2 says that "I cannot comment on the soundness of the concepts of "tresinos" and "Cooper pair formations" as this is not my field of expertise and these concepts are not explained in the manuscript." Knowing that this would present a problem to reviewers of this Brief Communication in my revised version of the paper, I suggested: In order to facilitate clearer understanding, the reader is urged to consider our first paper (Mayer and Reitz, 2014) before addressing the present Brief Communication. It is expected that most geophysicists are believers in the Standard Earth Energy Paradigm (SEEP) which we show to be inconsistent with most of the geophysics measurements of heat and helium generation from the Earth. Because of this, some effort is required to fully understand the present work.

Third, because Referee #2 stated: "I cannot comment on the soundness of the concepts of "tresinos" and "Cooper pair formations" as this is not my field of expertise and these concepts are not explained in the manuscript". Given Referee #2's own admission, it seems clear that he has been unwilling to spend the time to present a technically qualified review of my paper.

Fourth, Referee #2 says: "The only reference to a paper on magnetotellurics is cited out of context". It's not out of context because the referenced paper makes an important point that Magnetotelluric data scans often find zones of very-high electrical conductivity; see, for example, a typical paper by (Ritter, et al. , 1999). Furthermore, the authors of this paper comment "A large number of electrical conductivity anomalies have been detected in the Earth crust around the world". In addition, these authors say "there is no clear consensus as to the causes and origins of these anomalies, particularly in crystalline regimes". Referee #2 seems to believe that, in fact, there is consensus in the geophysics community. Just because there are carbon deposits does not mean that they are responsible for MT data at a basic physics level. Furthermore, adding more references to this and other MT processes does not resolve the issue of

the underlying physics of the MT observations.

Finally, I believe that the new proposal for the MT observations is important to consider because the same electron pair donors are required to understand the Earth's energy generation observed at many places around the Earth at relatively shallow depths as I discuss in this Brief Communication.

---

## Editor Comment (EC1) · R. Gloaguen (Editor) · 11 Jun 2018

I remind the author that the reviewers were selected by the handling editor and that there is a reason for this choice.

I also remind the author that antagonizing the reviewer does not help his case. The author is entering the field of deep Earth imaging without a priori knowledge and we need experts of that precise field. These experts are not familiar with cooper pairs. Neither is the handling editor.

Rev#2 did a very difficult job and I appreciate his effort.

[Figure]

2018-13, 2018.